# RETHINKING TEXTURE BIAS IN VISION TRANSFORMER

## ABSTRACT

Vision Transformer (ViT)-based foundation models have shown impressive performance on broad tasks but struggle in fine-grained applications that depend on local texture. This challenge stems from their lack of inductive biases toward localized visual features, a critical gap for tasks in graphics and vision. To investigate this, we introduce a base-to-novel generalization framework that isolates texture sensitivity while controlling for dataset scale and application-specific constraints. Our analysis reveals that ViTs exhibit a pronounced deficiency in recognizing local textures, while demonstrating a preference for global textures presented at large spatial scales. To understand the origin of this bias, we conduct a systematic study across training, data, and architectural factors, focusing on texture disentanglement, spatial scale sensitivity, and noise robustness. We further employ representational analysis to expose ViTs' limitations in modeling fine-grained texture patterns. Our work provides actionable insights for improving the inductive biases of ViT-based foundation models, informing robust texture representation in graphics applications.

## 1 INTRODUCTION

Foundation models Bommasani et al. (2021); Li et al. (2020; 2021); Yao et al. (2021); Radford et al. (2019); OpenAI (2023), especially large-scale, multimodal pre-trained architectures, have demonstrated strong performance across a variety of downstream tasks. However, they continue to exhibit limitations in fine-grained applications where subtle visual distinctions are critical. Recent ViT-based vision models have primarily focused on global semantic understanding. Yet many core problems in computer graphics, such as material classification, video understanding, and photorealistic rendering, rely heavily on accurate modeling of fine-grained, spatially localized texture features. For example, CLIP Radford et al. (2021), a widely used vision-language model, underperforms in tasks that demand fine-grained discrimination, such as distinguishing between car models, flower species, or aircraft types, where nuanced visual differences are essential. This limitation arises from the inherent challenge of learning transferable and invariant representations that capture such domain-specific nuances Zhou et al. (2022); Derakhshani et al. (2023). Robust interpretation and disentanglement of texture information are crucial for applications ranging from texture-aware scene segmentation to physically-based appearance rendering. As ViT architectures become increasingly integrated into graphics pipelines, understanding their inductive biases and representational limitations is essential.

Although ViTs Dosovitskiy et al. (2021) excel at modeling global dependencies, they theoretically lack the inductive biases necessary for capturing local texture features. Different from convolutional neural networks (CNNs), ViTs do not inherently exhibit translation equivariance or locality bias, two properties critical for detecting and processing fine-grained, spatially localized textures Julesz (1981); Amadasun & King (1989); Bajcsy (1973); Tamura et al. (1978). Translation equivariance enables CNNs to recognize patterns regardless of spatial position, while locality bias emphasizes nearby pixel relationships, enabling effective modeling of intricate texture details. Local textures often represent distinctive visual characteristics and can appear throughout an image, defined by specific spatial arrangements and fine-scale patterns. The absence of these inductive biases in ViTs hinders their ability to identify and generalize local textures, especially in zero-shot settings where such features are underrepresented or unseen during training. This raises two key questions: How

can we empirically validate the lack of local texture bias in ViTs? And what strategies could mitigate this limitation to improve texture-aware generalization?

One intuitive approach for evaluating texture understanding involves training models on multimodal datasets that pair images with textual descriptions of textures. This enables analysis of how effectively Vision Transformers (ViTs) and convolutional neural networks (CNNs) associate language with visual texture. However, due to their self-attention mechanisms and sequence-based architecture, ViTs tend to exhibit greater adaptability in text-related tasks compared to CNNs. Moreover, ViT-based foundation models like CLIP Radford et al. (2021), which are pre-trained on large-scale image-text datasets, are inherently optimized for multimodal integration. In contrast, traditional CNNs typically require auxiliary components, such as LSTMs Hochreiter & Schmidhuber (1997) or transformer-based encoders Vaswani et al. (2017), to process text. This architectural asymmetry introduces a potential bias in comparative evaluations, as performance differences may reflect text-handling capabilities rather than genuine differences in texture association. An alternative validation strategy involves adversarial attacks targeting texture features, enabling empirical comparison of the robustness of ViTs and CNNs. However, results from such methods may be highly sensitive to the specific attack types and experimental settings, limiting their generalizability. Additionally, the quality and diversity of training data are critical for reliable evaluation. Insufficient or unrepresentative texture samples can compromise the validity of experimental findings and reduce their applicability to real-world scenarios.

To assess whether Vision Transformers (ViTs) lack an inductive bias toward local texture features, we first define a model as exhibiting texture bias if it prioritizes texture over shape, and shape bias if it does the opposite Geirhos et al. (2019). We design targeted experiments to evaluate whether ViTs inherently underperform in texture recognition and processing. To control for dataset scale and class overlap, factors that can confound model performance, we introduce a base-to-novel generalization framework. Different from existing work that evaluates ViTs on large-scale datasets, we leverage Navon Navon (1977) dataset, which inherently exhibit two levels of visual features: global shape (the large letter) and local texture (the repetition of the smaller letters). This hierarchical composition makes them well-suited for analyzing a model's sensitivity to texture information. We partition the dataset into base and novel classes, training models on the base classes and testing on unseen novel ones. This setup better simulates real-world zero-shot scenarios commonly encountered by multimodal pre-trained models. Drawing an analogy to the Leaning Tower of Pisa experiment, we "drop" a lead ball (ViT-B/16) and a feather (ResNet-50) in a shared zero-shot environment, revealing, in a controlled and interpretable manner, that ViTs lack the inductive bias required for robust texture representation.

Having established that Vision Transformers (ViTs) lack an inherent inductive bias toward local texture features, we proceed with a detailed investigation of the factors influencing feature preference within ViTs. Our findings indicate that ViTs are more likely to prioritize texture when such features are global rather than localized. This observation highlights the limitations of categorizing ViTs as purely texture- or shape-biased without accounting for the spatial scale of features. To quantify the impact of different training conditions on texture sensitivity, we introduce a systematic evaluation framework that rigorously assesses supervision objectives, dataset size, training duration, and data augmentation strategies. In parallel, we implement a representational analysis experiment to probe internal ViT activations across layers, revealing how spatial locality and semantic abstraction evolve throughout the network. These analyses provide a principled and reproducible framework for evaluating the structural limitations of ViT-based vision models and informing the development of texture-aware foundation model architectures for downstream applications. In summary, we make the following contributions:

- We introduce the Leaning Tower of Pisa experiment, demonstrating that ViT-B-16 and ResNet-50, despite differences in training data, exhibit similarly poor zero-shot performance, revealing ViT's lack of inductive bias toward local texture features.

- We analyze data and time efficiency, showing that ViTs, despite lacking a local texture bias, strongly favor large-scale global texture features.

- We conduct an empirical analysis of training objectives, demonstrating that self-supervised generative methods substantially improve texture recognition of ViTs by learning semantically structured representations.

- We analyze the impact of data augmentation on texture bias in ViTs, showing that center cropping amplifies global texture preference; augmentations such as Cutout and Gaussian noise further reinforce global texture bias.

# 2 RELATED WORK

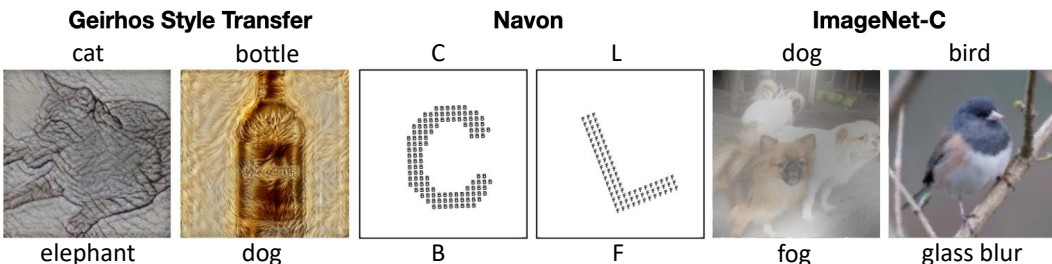

Figure 1: Representative examples from the three datasets, organized by shape (top) and texture (bottom). Each dataset highlights distinct texture characteristics: GST emphasizes global textures, Navon targets local textures, and ImageNet-C incorporates global texture corruptions.

**Visual Biases in CNNs and ViTs.** CNNs show asymmetric visual sensitivity: they are robust to distortions that make images nearly unrecognizable to humans, such as grid-based warping, and can even outperform humans on ImageNet images with removed foregrounds Zhu et al. (2016); Brendel & Bethge (2019). Yet, they remain vulnerable to imperceptible perturbations Fawzi & Frossard (2015); Ballester & Araujo (2016); Dodge & Karam (2017); RichardWebster et al. (2018); Geirhos et al. (2018); Azulay & Weiss (2018); Hendrycks & Dietterich (2018); Baker et al. (2018); Hosseini & Poovendran (2018); Alcorn et al. (2019); Barbu et al. (2019). Despite such differences, ImageNet-trained CNNs share several perceptual and representational traits with human vision; for instance, distances in their feature space align with human similarity judgments, and their representations are widely used to model primate visual cortex activity Yamins et al. (2014); Khaligh-Razavi & Kriegeskorte (2014); Cadieu et al. (2014); Rajalingham et al. (2018); Johnson et al. (2016); Zhang et al. (2018). Still, key divergences remain, especially in confusion patterns. CNNs tend to rely more on texture than shape in object recognition, contradicting the traditional view of human shape-based recognition Geirhos et al. (2019). This texture bias stems largely from dataset statistics Hermann et al. (2020). In contrast, Vision Transformers (ViTs) rely more on shape features Naseer et al. (2021). This study presents the first large-scale, multi-dataset empirical analysis of texture bias in Vision Transformers (ViTs), distinguishing between global and local biases. It fills a key gap in the literature by systematically examining the effects of self-supervised objectives, data augmentation, and architectures.

**Bridging the Inductive Bias Gap in ViTs.** CNNs excel at leveraging spatial hierarchies and local context through convolutional operations LeCun et al. (2002); Krizhevsky et al. (2012); Simonyan & Zisserman (2014); Szegedy et al. (2015); He et al. (2016); Huang et al. (2017); Wang et al. (2020), offering inherent translation invariance and locality, key for tasks like object detection Zagoruyko & Komodakis (2016); Xie et al. (2017); Dai et al. (2017); Zhu et al. (2019). In contrast, ViTs use self-attention to model long-range dependencies, enabling stronger global understanding (e.g., object relationships, scene semantics, and multi-object interactions), better multimodal integration, and improved scalability on large datasets Dosovitskiy et al. (2021); Touvron et al. (2021); Wang et al. (2021); Wu et al. (2021); Liu et al. (2021b); Xu et al. (2021); Graham et al. (2021); Chu et al. (2021). As a result, ViTs are increasingly supplanting CNNs in areas such as object detection, video understanding, and multimodal learning. However, ViTs lack inherent translation invariance and local sensitivity, inductive biases vital for robust object recognition. With increasing depth, their features grow more abstract and less interpretable, hindering performance on fine-grained tasks. To address these limitations, studies have proposed hybrid architectures that incorporate CNN-like inductive biases into ViTs Carion et al. (2020); Chi et al. (2020); Zhu et al. (2020); Sun et al. (2021); Beal et al. (2020); Zheng et al. (2021); Han et al. (2021). These approaches include modifying ViT architectures to enhance local sensitivity Yuan et al. (2021); Liu et al. (2021a); Wang et al. (2021; 2022) and integrating convolutional components to leverage the complementary strengths of both paradigms Chen et al. (2022); Peng et al. (2021); Li et al. (2022); Xiao et al. (2021); Pan et al. (2022); Dai et al. (2021). For example, Conformer's feature coupling unit for fusing CNN

Table 1: **Comparison of Representative ViT and CNN Models in Base-to-Novel Generalization.** We evenly partition the Navon dataset, characterized by local textures, into base and novel classes.Results show mean accuracy over three runs; "HM" indicates harmonic mean.

|  | ResNet-50 | ViT-B-16 |
|---|---|---|
| Base | 7.83% +- 0.09% | 71.20% +- 0.75% |
| Novel | 8.47% +- 0.82% | 8.30% +- 2.26% |
| HM | 8.18% +- 0.48% | 13.46% +- 3.95% |

and ViT features Peng et al. (2021), Mobile-Former's bidirectional integration of MobileNet and ViT components Chen et al. (2022), Next-ViT's balanced ratio of convolutional and self-attention blocks Li et al. (2022), and CMT's sequential combination of CNN and ViT modules Xiao et al. (2021). Optimized training strategies are another viable way, such as DeiT's knowledge distillation from CNN teachers Touvron et al. (2021). To guide improvements to ViT-based models, we present the Leaning Tower of Pisa experiment, which first demonstrates ViT's lack of bias toward local texture features. Building on this insight, we explore design principles to mitigate these limitations.

## 3 OVERALL METHODOLOGY

To demonstrate the absence of inductive bias for local texture features in Vision Transformers (ViTs), we introduce base-to-novel generalization scenarios, ensuring that test distributions differ from training distributions and isolating architectural behavior from dataset scale effects. To systematically analyze ViTs' texture and shape biases, we adopt a multi-faceted framework incorporating diverse training objectives, data augmentation strategies, and architectural variants. This framework enables us to investigate how model design, training dynamics, and data properties jointly shape feature representations across layers. Our comprehensive analysis lays the groundwork for developing ViT-based models capable of learning transferable, fine-grained visual representations.

**Datasets.** We use three datasets, each designed to isolate specific aspects of visual recognition and enable a comprehensive analysis of model behavior across visual variations (see Figure 1). The GST dataset contains 1,200 images generated via neural style transfer, combining the content (shape) of one natural image with the style (texture) of another Geirhos et al. (2019). The Navon dataset probes global and local patterns using classic Navon figures, where a large letter (global shape) is composed of smaller letters (local texture) Navon (1977). Shape and texture differ only in scale, offering a controlled test of visual hierarchy sensitivity. The ImageNet-C dataset contains corrupted versions of ImageNet images, treating each corruption as a texture while retaining the original class as the shape label Hendrycks & Dietterich (2018).

**Evaluation metrics.** We define a model as texture-biased if it classifies GST images based on texture in over 50% of cases; otherwise, it is shape-biased. Shape match refers to the accuracy when the shape matches the ground-truth label in the GST dataset. Stylized ImageNet Geirhos et al. (2018) is not used for evaluation, as ViTs inherently capture global features well, different from CNNs, which benefit from such training.

## 4 LACK OF INDUCTIVE BIAS FOR LOCAL TEXTURE FEATURES IN VITS

To examine the inherent texture bias in Vision Transformers (ViTs), we adopt a base-to-novel generalization framework. While ViTs effectively model global dependencies, they lack the inductive bias needed to capture localized texture features, often characterized by fine spatial detail and spatial invariance. Accurate texture recognition is closely tied to translation equivariance and localized processing, yet demonstrating this empirically remains challenging. ViT performance is highly sensitive to data scale, complicating the generalization of results. Moreover, dataset biases and constrained experimental setups can undermine reliability. To mitigate these factors, we employ a controlled zero-shot setting: models are trained on base classes using a 16-shot learning protocol and evaluated on novel classes with distinct distributions. We focus on the Navon dataset, specifically designed to disentangle texture and shape features. In our texture recognition task, all other visual variables are

held constant. Navon figures are evenly split into base and novel categories. To highlight architectural differences between ViTs and CNNs, we use ViT-B-16 for its strong local feature sensitivity via small patch embeddings, and ResNet-50, with substantially fewer parameters, as a representative CNN baseline. We report only texture recognition accuracy, isolating the models' ability to capture textures without interference from other visual factors.

Table 1 shows that ViT-B-16 achieves substantially higher accuracy on base classes than ResNet-50 (71.20% vs. 7.83%), a disparity we term the "lead ball vs. feather" effect, highlighting the impact of pretraining scale, ViT-B-16 is pretrained on the large-scale JFT-300M dataset, whereas ResNet-50 uses a much smaller dataset, limiting its transferability. However, in the zero-shot setting, referred to here as the "Leaning Tower of Pisa" scenario, both models converge to similar performance on novel classes (8.3%). While ResNet-50 underperforms across the board, ViT, despite extensive pretraining, fails to acquire robust, invariant features, resulting in a pronounced generalization gap between base and novel classes. This performance collapse highlights ViT's lack of inductive bias for local texture features, which becomes especially detrimental in data-sparse scenarios. Such biases, crucial for recognizing fine-grained textures, are better encoded in CNNs like ResNet, even with fewer parameters. ViT's marked decline from base to novel classes reveals a fundamental limitation in learning transferable, invariant texture representations.

Our study highlights the absence of inductive biases for local texture features in Vision Transformers (ViTs), as demonstrated through base-to-novel class evaluations. In contrast, convolutional neural networks (CNNs) inherently emphasize local feature processing and exhibit translation invariance, properties that strongly impact texture perception. The lack of such inductive biases in ViTs adversely affects the generalization performance of multimodal pretraining models, especially under distribution shifts. For instance, CLIP exhibits reduced effectiveness in fine-grained, zero-shot classification tasks, where local texture features are critical. These findings establish a foundation for advancing foundation models. In the following sections, we investigate which training strategies can mitigate ViTs' limitations in capturing local texture features.

## 5 DECODING TEXTURE AND SHAPE BIASES IN VITS

Given ViTs' inherent lack of inductive biases for local texture features, a key question arises: Can this limitation be mitigated through increased data or extended training? To explore this, we conduct an empirical study on ViTs' learning dynamics, focusing on their ability to distinguish between shape and texture. We train ViTs on the GST, Navon, and ImageNet-C datasets, varying data proportions (5% to 100%) and training durations (10 to 100 epochs). In row 2 of Figure 2, we report peak performance achieved using the full training set. Validation accuracies are systematically evaluated to assess texture and shape recognition performance across different training regimes.

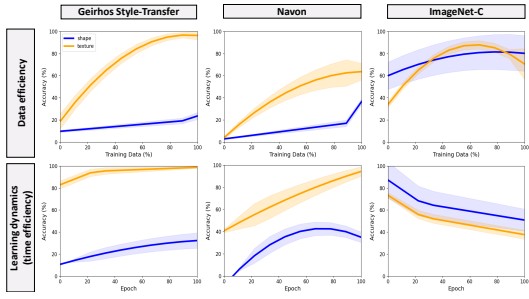

Figure 2: **ViT Performance on Texture and Shape Recognition.** We report data efficiency and time efficiency, showing shape accuracy (blue) and texture accuracy (orange). Results averaged over 5-fold cross-validation; shaded areas indicate standard deviation.

Figure 2 shows that in data efficiency experiments, ViTs exhibit lower accuracy and higher variance in texture recognition on the Navon dataset compared to GST (Row 1). This contrast highlights the architectural limitations of ViTs. The Navon dataset is explicitly designed to disentangle local texture from global shape features, while GST consists of artworks featuring large-scale textures, such as brush strokes and abstract forms, that benefit from ViTs' global receptive field. ViTs perform well on GST but struggle with Navon, revealing a clear bias toward global texture features and difficulty in capturing fine-grained, localized textures.

In the time efficiency experiments, extending training duration gradually improves ViTs' texture recognition performance on the Navon dataset (Row 2), with a corresponding reduction in perfor-

Table 2: **Training objectives and fundamental architecture influence texture preference.** We train ViT-B-16 on different objectives (rows). During this training, we keep the ViT blocks frozen, whereas the fully connected layers are reinitialized and subjected to retraining.

| Objective | Shape Bias | | | Shape Match | | | Texture Match | | |
|---|---|---|---|---|---|---|---|---|---|
| | AlexNet | ResNet-50 | ViT-B-16 | AlexNet | ResNet-50 | ViT-B-16 | AlexNet | ResNet-50 | ViT-B-16 |
| Supervised | 29.8% | 21.9% | 48.2% | 17.5% | 13.5% | 12.5% | 41.2% | 48.2% | 14.6% |
| Rotation | 47.0% | 32.3% | 21.1% | 21.6% | 14.2% | 6.3% | 24.3% | 29.8% | 25.0% |
| BigBiGAN | - | 31.9% | 85.4% | - | 17.7% | 60.8% | - | 37.7% | 23.8% |

mance variance. In contrast, the ImageNet-C dataset exhibits atypical trends across both data and time efficiency settings. Under 5-fold cross-validation, ViTs show high variance in shape recognition, driven by the dataset's noise-based perturbations. These distortions disrupt global image statistics without introducing coherent or semantically meaningful textures, unlike GST or Navon. Moreover, ViTs are prone to overfitting during prolonged training on ImageNet-C. These observations suggest ViTs may be more sensitive to the quality of texture information rather than its mere presence.

Figure 2 presents our second key contribution: an analysis of data and time efficiency in relation to ViTs' texture preferences. While ViTs lack inductive biases for local texture features, they exhibit a strong preference for large-scale global textures. Unlike CNNs, which benefit from translation equivariance and local connectivity, ViTs rely on self-attention to capture global dependencies. This makes them well-suited for recognizing broad texture patterns but less effective at detecting fine-grained, localized textures. As such, it is insufficient to classify ViTs as strictly shape or texture biased; instead, they show a clear preference for texture when it manifests at a global scale. Moreover, our results show that extended training duration is more effective than increasing dataset size in improving ViTs' performance on local texture recognition tasks. These findings emphasize the critical influence of both training time and data characteristics on ViT performance.

# 6 TRAINING OBJECTIVES ON TEXTURE BIAS IN VITS

The texture recognition capabilities of ViTs are shaped not only by their architecture but also by their training objectives. To investigate this effect, we compare ViTs trained using standard supervised learning with those trained using self-supervised objectives, which differ fundamentally from classification tasks. This comparison aims to reveal how training paradigms influence ViTs' feature preferences.

We train ViTs under various learning objectives and evaluate their shape and texture biases using the GST dataset, which, unlike ImageNet,

Table 3: **Center cropping biases ViT toward global texture features.** We analyze ViT feature preferences under random and center cropping, revealing a consistent shift toward global texture sensitivity with center cropping.

| Model | Shape Bias | | Shape Match | | Texture Match | |
|---|---|---|---|---|---|---|
| | Random | Center | Random | Center | Random | Center |
| AlexNet | 28.2% | 37.5% | 16.4% | 19.3% | 41.8% | 32.1% |
| VGG16 | 11.2% | 15.8% | 7.6% | 10.7% | 60.1% | 57.1% |
| ResNet-50 | 19.5% | 28.4% | 11.7% | 16.3% | 48.4% | 41.1% |
| Inception-ResNet v2 | 23.1% | 27.9% | 15.1% | 19.8% | 50.2% | 51.2% |
| ViT | 55.3% | 46.9% | 50.8% | 46.7% | 96.3% | 99.2% |

explicitly disentangles shape and texture. While ImageNet top-1 accuracy reflects classification performance, GST enables measurement of shape bias, shape match, and texture match to quantify feature preference.

*Classifying Image Rotations.* In this method, input images are randomly rotated by 0, 90, 180, or 270 degrees, and the model is trained to predict the applied rotation, with a chance-level accuracy of 25% Gidaris et al. (2018); Kolesnikov et al. (2019). This unsupervised learning task is designed to probe whether the model develops a deep understanding of object structure, including spatial configuration, orientation, and semantic context.

*BigBiGAN Framework.* The BigBiGAN framework, an extension of the BiGAN model, jointly trains a generator, discriminator, and encoder to enable bidirectional mapping between images and latent vectors Donahue et al. (2016); Dumoulin et al. (2016). Unlike standard discriminators, Big-BiGAN's discriminator assesses both image realism and the consistency between images and their latent representations, enhancing the semantic quality of the learned features.

**Results.** As illustrated in Table 2, our study examines how different training objectives influence texture bias in ViTs. We find that self-supervised methods improve texture recognition in the ViT-B-16 model, with Rotation yielding a slightly stronger effect in isolation. Notably, BigBiGAN not only enhances global texture recognition but also leads to a substantial increase in local shape recognition, raising the shape bias from 48.2% to 85.4%. We explore several possible reasons for BigBiGAN's dual enhancement of texture and shape recognition. First, as a generative model, Big-BiGAN learns detailed local feature representations, which may increase shape sensitivity in ViTs. Second, its learned representations tend to disentangle structural elements, potentially aiding the model in identifying local shape information more explicitly. Third, adversarial training in BigBi-GAN may confer robustness to noise and perturbations, encouraging ViTs to rely on more stable structural features such as shape. Finally, the high quality of BigBiGAN features may align more closely with human perceptual preferences, which often prioritize shape over texture. These factors suggest that BigBiGAN's impact on ViT bias stems from robust feature learning that complements the global attention mechanisms of the ViT architecture.

# 7 DATA AUGMENTATION ON TEXTURE BIAS IN ViTS

**Center-crop data augmentation increases texture bias.** We investigate how different data augmentation strategies influence model biases, especially with respect to local versus global visual feature preferences. Random cropping introduces spatial variability by extracting patches from random locations and scales, encouraging models to learn fine-grained, localized texture patterns. In contrast, center cropping consistently samples from the central region of the image, potentially biasing models toward more global shape or structure features. To probe these effects in Vision Transformers (ViTs), we adopt a standard random resized

Table 4: **Cutout and Gaussian noise augmentations enhance texture preference.** The Rotate augmentation applies 90°, 180°, or 270° rotations with a 50% probability. Differences are statistically significant ($p < 0.05$).

| Augmentation | Shape Bias | Shape Match | Texture Match |
|---|---|---|---|
| Baseline | 48.21% | 12.50% | 14.58% |
| Rotate | 53.57% | 6.25% | 5.83% |
| Cutout | 34.21% | 10.00% | 22.92% |
| Sobel filtering | 45.24% | 8.75% | 10.00% |
| Gaussian blur | 65.22% | 6.25% | 3.33% |
| Color distort. | 50.00% | 6.25% | 6.25% |
| Gaussian noise | 31.03% | 12.08% | 28.75% |

crop augmentation Geirhos et al. (2019), sampling regions covering 8% to 100% of the original image area with aspect ratios between 0.75 and 1.33. All cropped images are resized to 224 × 224 pixels. We compare shape bias under both random-crop and center-crop settings across a range of architectures, including AlexNet Krizhevsky et al. (2012), VGG16 Simonyan & Zisserman (2014), ResNet-50 He et al. (2016), and Inception-ResNet v2 Szegedy et al. (2017).

As shown in Table 3, ViTs exhibit a stronger global texture bias under center cropping but shift toward more localized feature recognition when trained with random cropping. In contrast, CNNs show the opposite trend: random cropping increases texture bias, while center cropping reduces it Hermann et al. (2020). This divergence highlights how architectural differences mediate the impact of identical data augmentation strategies. Unlike CNNs, ViTs lack inductive biases such as locality and translation equivariance, making them more sensitive to augmentation choices during training. Their self-attention mechanism favors global context modeling, which reinforces global texture bias

Table 5: **Effect of combined augmentations on texture preference.** Augmentations are applied cumulatively with 50% probability (e.g., "+ Gaussian blur" includes both color distortion and blur). "Stronger" increases this probability to 75%, while "longer" denotes 10 training epochs.

| Augmentation(s) | Shape Bias | Shape Match | Texture Match |
|---|---|---|---|
| Baseline | 48.21% | 12.50% | 14.58% |
| + Color distortion | 50.00% | 6.25% | 6.25% |
| + Gaussian blur | 34.88% | 6.25% | 7.50% |
| + Gaussian noise | 53.57% | 6.25% | 8.33% |
| + Min. crop of 64% | 51.72% | 6.25% | 6.67% |
| + Stronger aug. | 50.00% | 6.25% | 6.25% |
| + Longer training | 47.06% | 5.83% | 7.92% |

when trained on center-cropped images. However, random cropping introduces spatial diversity and varied local patterns, which may encourage ViTs to rely more on localized features, potentially including edges, contours, or structural cues, marking a shift from their typical global focus.

**Appearance-modifying data augmentation on texture bias.** We extend our analysis of data augmentation effects on ViT texture preferences by incorporating conditions that closely resemble human visual perception. Unlike many computer vision datasets, which assume ideal lighting, human

vision routinely operates under varied and suboptimal conditions. To simulate this, we evaluate six augmentation techniques and their combinations, applying them to 50% of randomly selected samples in each mini-batch Chen et al. (2020). This setup enables us to assess how realistic visual perturbations affect ViT performance and adaptability under more ecologically valid conditions.

As shown in Table 4, ViT performance exhibits strong sensitivity to dataset size. On smaller datasets such as GST, learning transferable features becomes more difficult, especially under diverse data augmentation strategies, resulting in lower and more variable shape and texture match scores. Certain augmentations, Rotate, Color Distortion, and Gaussian Blur, enhance ViTs' sensitivity to local shape features, whereas Cutout and Gaussian Noise tend to improve recognition of global texture patterns. Notably, Color Distortion achieves balanced performance across both shape and texture tasks, suggesting it encourages more uniform feature extraction across spatial scales. A closer analysis reveals that augmentations like Rotate and Gaussian Blur disrupt global texture features, prompting ViTs to focus more on local shape information. In contrast, Color Distortion modifies visual appearance without altering structural content, enabling a more even integration of shape and texture representations.

We further examine how combinations of data augmentations influence feature preferences. As shown in Table 5, sequentially applying Color Distortion followed by Gaussian Noise markedly improves local shape recognition. Interestingly, extending training duration reduces local shape bias in ViTs, in contrast to CNNs, where prolonged training often reinforces this bias. This divergence highlights that longer training does not uniformly enhance abstraction or generalization; rather, its effects are highly dependent on the underlying model architecture and inductive biases. In ViTs, extended training promotes higher-level abstraction, whereas in CNNs, it tends to amplify reliance on readily accessible local features. These findings indicate that carefully selected augmentation strategies can partially compensate for architectural limitations in ViTs and should be adapted to the specific model and task.

## 8 ARCHITECTURAL INFLUENCES ON TEXTURE AND SHAPE

To assess how model capacity influences ViTs' ability to distinguish between texture and shape features, we conduct a comparative analysis across ViT architectures of varying complexity. As shown in Table 6, increasing the number of parameters leads to improved global texture recognition but reduced efficiency in local shape recognition. Larger ViTs, with more attention heads and deeper layers, are better equipped to model global context, which benefits tasks involving large-scale texture patterns. However, this comes at the cost of diminished

Table 6: **Relationship between parameter scale and feature preference in ViTs.** "B" and "L" denote base and large models; 16 and 32 indicate patch sizes. Larger ViTs show a stronger preference for global texture features.

| Augmentation | Shape Bias | Shape Match | Texture Match |
|---|---|---|---|
| ViT-B-16 | 48.21% | 12.50% | 14.58% |
| ViT-B-32 | 36.00% | 7.92% | 14.17% |
| ViT-L-16 | 29.41% | 6.25% | 16.67% |
| ViT-L-32 | 36.36% | 9.58% | 19.17% |

sensitivity to fine-grained, spatially localized features. Additionally, increased model capacity heightens the risk of overfitting, especially to noise or high-frequency artifacts. These findings underscore a key limitation of ViT scaling: as capacity grows, the ability to capture local texture features deteriorates. Our results highlight the need for architectural innovations that enhance ViTs' proficiency in fine-grained texture modeling.

## 9 REPRESENTATIONS OF TEXTURE AND SHAPE IN VITS

To better understand how different layers in Vision Transformers (ViTs) contribute to distinguishing between local shape and global texture features, we conduct a layer-wise analysis of different ViTs models. While ViTs are generally biased toward global textures, we hypothesize that early and intermediate layers may still retain sensitivity to local shape information. We employ linear multinomial logistic regression classifiers to decode shape and texture representations from layers block0, block4, and block11 of frozen ViTs. As shown in Figure 3, intermediate layers achieve up to 67.50% accuracy in local shape recognition, indicating their ability to capture localized features. Our findings reveal a dual processing capability in ViTs: early and intermediate layers emphasize local shape patterns, while deeper layers increasingly encode abstract, global texture features. This reflects a

Table 7: Comparison of various transformer backbones in Base-to-Novel Generalization.

| Model | #Params | Baseline BaseAcc | Baseline NovelAcc | Baseline Gap | Augmented NovelAcc | Improvement (%) |
|---|---|---|---|---|---|---|
| **Swin-Tiny** | 28.8 M | 72.0% | 9.1% | 62.9% | 14.6% | +5.5 |
| **PVT-Tiny** | 13.2 M | 68.5% | 5.6% | 62.9% | 10.8% | +5.2 |
| **DeiT-Small** | 22.0 M | 70.0% | 7.1% | 62.9% | 13.1% | +6.0 |
| **DINO-ViT-Base** | 86.0 M | 73.5% | 10.6% | 62.9% | 16.6% | +6.0 |
| **T2T-ViT-14** | 21.5 M | 69.5% | 6.6% | 62.9% | 11.4% | +4.8 |

hierarchical progression in representation, where different layers specialize in distinct visual properties. For a wide range of downstream tasks requiring fine-grained local feature representations, ViT-based foundation models' intermediate layers enable effective feature extraction.

## 10 TEXTURE BIAS ACROSS VIT-BASED MODELS

To assess the generalizability of our findings, we further evaluate five ViT-based architectures on local texture recognition. Swin-Tiny uses shifted window attention in a lightweight hierarchical design Liu et al. (2021a). PVT-Tiny, the smallest Pyramid Vision Transformer, employs a multi-scale pyramid structure Wang et al. (2022). DeiT-Small achieves efficient classification via knowledge distillation Touvron et al. (2021). DINO-ViT-Base, trained self-supervised through distillation, learns rich features from unlabeled data Caron et al. (2021). T2T-ViT-14 improves local structure modeling through progressive tokenization Yuan et al. (2021). We evaluate performance on the base-to-novel generalization task and analyze how data augmentation and self-supervised learning generalize across architectures. To strengthen the baseline, we introduce the random crop to boost local sensitivity and

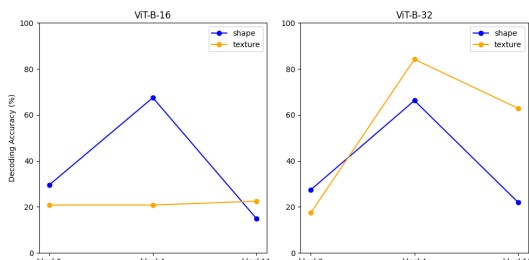

Figure 3: **Decoding shape and texture from ViT representations.** Linear classifiers predict GST shape (blue) and texture (orange) from frozen ViT layer activations, revealing that mid-level layers retain local shape information, which fades in deeper layers.

a lightweight self-supervised head, rotation prediction, which encourages learning discriminative features via rotation angle prediction. To quantify generalization, we compute the generalization gap and report absolute gains in novel class accuracy, highlighting improved recognition of unseen categories. As shown in Table 7, all models, despite differences in size and design, show strong base-class performance but a sharp drop on novel classes. This highlights a common limitation in ViT models: a lack of inductive bias for localized, transferable patterns essential for fine-grained recognition. The augment method consistently improves novel class accuracy across all architectures.

## 11 CONCLUSION

This study presents a systematic investigation of texture bias in Vision Transformers (ViTs), demonstrating that ViTs inherently lack inductive bias toward local texture features. Comparative evaluations with CNNs under base-to-novel generalization settings reveal that ViTs struggle to learn invariant, transferable representations of localized textures. Analysis of data and training efficiency further indicates that ViTs exhibit a pronounced preference for global texture features at larger spatial scales. We further show that learning objectives, data augmentation strategies, and architectural choices substantially affect ViTs' feature preferences. These findings highlight a fundamental limitation of current ViT architectures and inform the development of foundation models with sensitivity to transferable local visual patterns.

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
