# Rethinking Texture Bias in Vision Transformer
## —Supplementary Material—

## Spatial attention patterns in ViT representations

To further investigate the spatial attention behavior of Vision Transformers in fine-grained scenarios, we visualize attention weights from the class token to all patch tokens in the final layer of ViT-B-16. As shown in Figure 1, each column displays a Navon character stimulus, composed of repeated micro-letters forming a larger shape, alongside its corresponding class token attention heatmap. Across all stimuli, the attention patterns appear diffuse and low in magnitude, exhibiting minimal spatial concentration regardless of the global structure or orientation of the input. This suggests that the class token fails to selectively attend to spatially localized regions, even at the final representational stage.

Notably, the attention maps show little to no alignment with the local textures (i.e., repeated local letters), indicating a lack of spatial hierarchy or part-whole awareness. This reinforces a core architectural limitation of ViTs: in the absence of convolutional priors or hierarchical design, the model struggles to capture localized visual regularities critical for compositional or fine-grained recognition. The uniformly low attention responses and absence of discriminative peaks imply that final predictions are not grounded in spatially focused evidence, but are instead distributed across the image, consistent with prior observations of ViT's weak inductive bias toward locality.

These visualizations corroborate the quantitative performance drop observed in zero-shot generalization to novel classes, particularly in tasks where fine-grained, local texture features are essential. They further motivate the development of training strategies or architectural modifications that promote spatial selectivity and more interpretable attention dynamics in ViT-based models.

## Methodology

Due to the significant computational resources and time required, researchers and practitioners prefer to use pre-trained ViT models and fine-tune them on specific tasks or smaller datasets. This approach is generally more feasible and still leverages the powerful representational capabilities of ViT models. Our methodology adheres to this established practice, using pre-trained ViT models for all experiments detailed in the approach section. We fine-tune these models on selected datasets and directly assess the effects of diverse data augmentation techniques on model performance when testing on the ImageNet dataset. This approach efficiently facilitates the quick determination of the impact that varying training objectives have on ViT performance with fewer training epochs.

## Base-to-novel generalization

In the case of the Navon dataset, we generate three distinct training splits. Each split includes a random sampling of 16 instances from each image category. We divide the total categories into two equal parts, designated as base classes and novel classes. The model's performance is then assessed using the respective validation splits for each category. We present the accuracies for both base and novel classes, along with their harmonic mean (HM), averaged across three iterations. To guarantee a balanced comparison between the two architectures in the base-to-novel generalization experiment, identical hyperparameters are selected. The ResNet-50 and ViT-B-16 models underwent training over 10 epochs, each with a consistent learning rate of 1e-4.

## Learning experiments

**Dataset considerations**    Each dataset used in our study has limitations in accurately representing texture. For instance, in the GST dataset, human subjects struggled with the style classification task (achieving an average accuracy of 14.2% against a chance level of 6.25%) (Geirhos et al. 2019a), based on data from a human experiment with texture-biased instructions (originally shown in Fig 10b of plotted by shape class (Geirhos et al. 2019a); data sourced from (Geirhos et al. 2019b)). Additionally, variability arises from the performance of the style transfer algorithm on individual images, and the reliance of style transfer on ImageNet-trained CNN features means that the dataset isn't entirely independent of the models under evaluation. Furthermore, the noise textures in ImageNet-C are perhaps the least representative of typical textures as generally understood. We anticipate that presenting results across all three datasets will mitigate any unique quirks of each dataset. In our upcoming research endeavors, we aim to develop new datasets that merge the manipulability of Navon stimuli with the realistic qualities of the GST and ImageNet-C datasets.

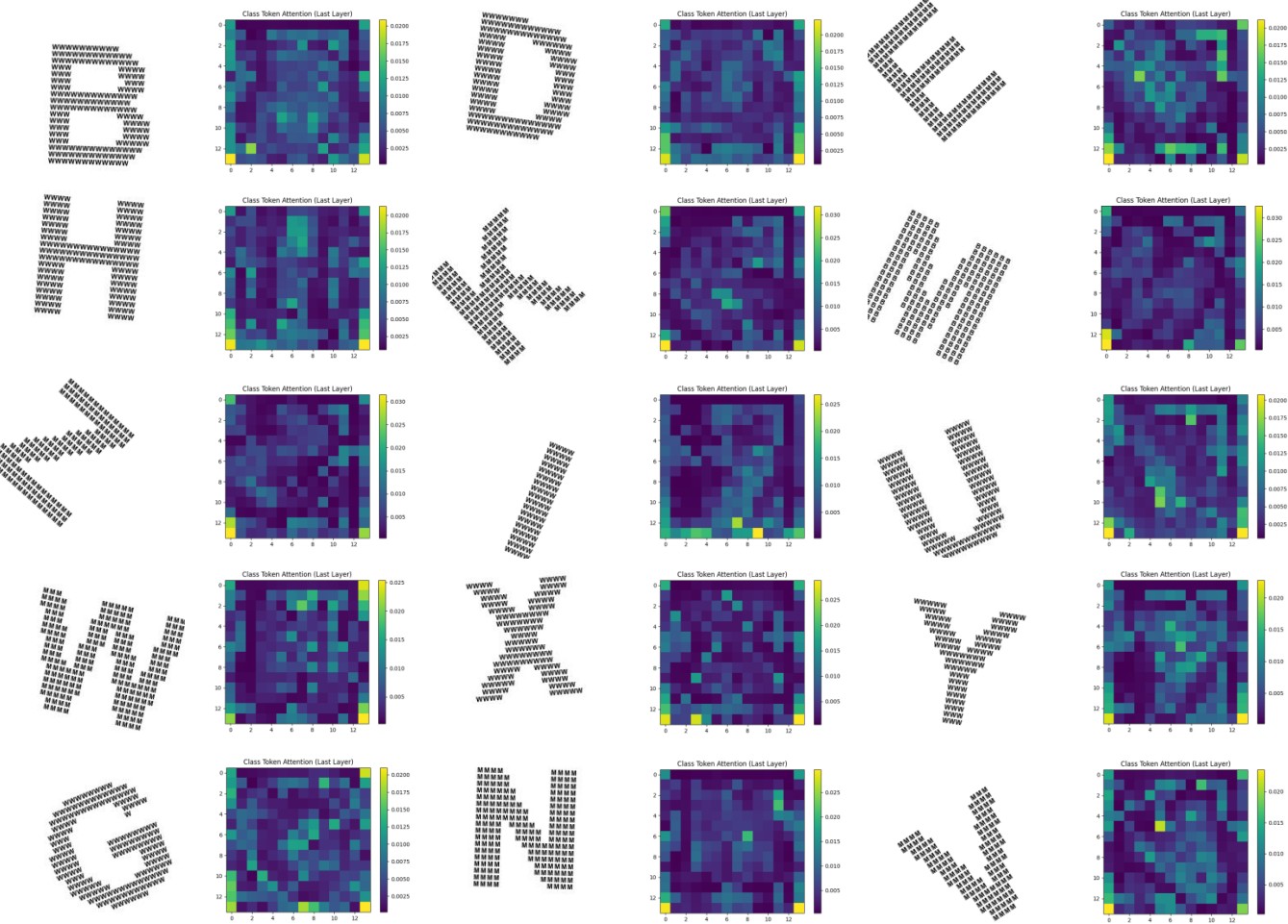

Figure 1: Attention maps from the final layer of ViT-B/16 on Navon stimuli. Each pair shows a composite character image (left), where a large letter is formed from repeated small letters, and the corresponding attention heatmap (right). The maps show diffuse, low-magnitude responses with no strong spatial selectivity or alignment to global or local structure. This illustrates the model's limited inductive bias for hierarchical or localized visual patterns, contributing to poor generalization in fine-grained tasks.

**Dataset splits**    *Geirhos Style-Transfer (GST) dataset*. We generate five cross-validation splits for the dataset, applying each split to both classification tasks. In creating a split, we reserve one exemplar each for shape and texture, ensuring that no entire shape or texture classes are omitted. Consequently, during the texture task, a model has to generalize across different exemplars of the same texture, and similarly, for the shape task, it needs to generalize across different exemplars of the same shape. The average validation set size across these splits is 483 items, accounting for approximately 40.3% of the total data. It's important to note that the dataset includes 80 images where shape and texture coincide. Following the precedent set by (Geirhos et al. 2019a), which excludes these images in calculating shape and texture biases, we also omit them from our analysis.

*Navon dataset*. In the case of the Navon dataset, we independently establish five cross-validation (cv) splits for each

specific task. In the shape task, we exclude three texture classes (such as the letters "W", "D", "K"), while for the texture task, three shape classes are set aside. The size of the validation set for each split was 375 items, which represents 11.5% of the total dataset.

*ImageNet-C dataset*. For each version of the dataset, we conduct separate splits for both shape and texture tasks. In the shape task, we exclude two texture classes (like "brightness" and "saturate"), while in the texture task, two shape classes are omitted (for example, brightness's "n03014705" and "n02098286"). This resulted in a validation set comprising 9,500 items, which constitutes 10.5% of the entire dataset.

**Training**    For both training and validation images, we apply preprocessing by adjusting the pixel values according to the mean and standard deviation of the training data subset. In the case of the GST and ImageNet-C datasets, each training im-

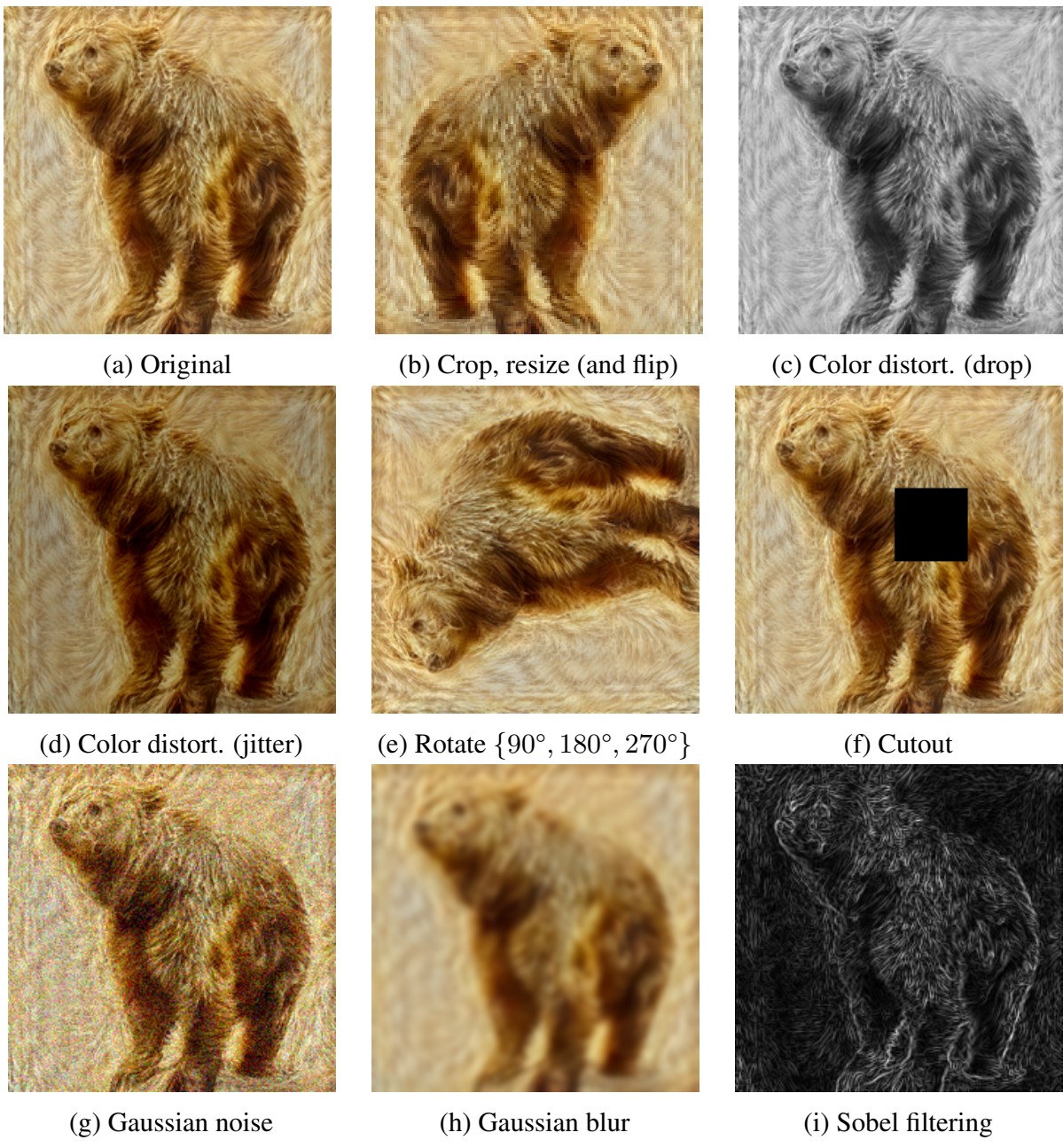

<table>
<tr><td>(a) Original</td><td>(b) Crop, resize (and flip)</td><td>(c) Color distort. (drop)</td></tr>
<tr><td>(d) Color distort. (jitter)</td><td>(e) Rotate {90°, 180°, 270°}</td><td>(f) Cutout</td></tr>
<tr><td>(g) Gaussian noise</td><td>(h) Gaussian blur</td><td>(i) Sobel filtering</td></tr>
</table>

Figure 2: Visual representations of the data augmentation techniques examined in the study. Each augmentation method stochastically alters data using specific internal parameters, such as the degree of rotation or the level of noise.

age is subjected to random horizontal flips with a probability of 0.5 during the training phase.

When subsampling the training data, we ensure that every shape and texture class is represented at least once. Specifically for the GST dataset, we reserve one exemplar from each texture and shape category in every split. For other datasets, shape classes are excluded in the texture training and vice versa.

**Evaluation of shape bias, shape match, and texture match**

For assessing shape and texture match, as well as shape bias in models trained on ImageNet, we follow (Geirhos et al. 2019a) where models are shown complete, non-cropped images from the GST dataset. We then collect the class probabilities generated by the model and aggregate them into 16 superclasses by summing the probabilities of the ImageNet classes within each superclass (Geirhos et al. 2019a). The

shape match metric is calculated as the frequency with which a model accurately identifies the shapes of the probe items. Texture match refers to the model's accuracy in predicting the textures of these items, and shape bias is determined as the proportion of instances where the model correctly identifies shape in cases where either shape or texture prediction is correct.

## Data augmentation and self-supervised representation experiments

For experiments detailed in Section 3.3 (Tables 3 and 4), random-flip augmentation is employed. Furthermore, we explore the impact of various other augmentations, including color distortion (comprising 80% probability of color jitter and 20% probability of color drop), rotation, cutout, Gaussian noise, Gaussian blur (with a kernel size amounting to 10% of the image's width/height), and Sobel filtering, as outlined in the study by (Chen et al. 2020). Except where specifically indicated, these augmentations are applied to roughly half of the examples (50% probability) in each mini-batch. Figure 2 features visual examples of these augmentations.

For experiments in Section 3.4 (Tables 5), all classifiers in our study are accompanied by data augmentation techniques that include random flipping and cropping. For the cropping, images were first resized to have their shortest side be 256 pixels, followed by cropping out regions of $224 \times 224$ pixels. This milder approach to cropping has been employed in prior research for training classifiers on self-supervised representations (Gidaris, Singh, and Komodakis 2018; Kolesnikov, Zhai, and Beyer 2019; Donahue and Simonyan 2019), and we find it crucial for replicating their results.

## Texture bias across ViT-based pre-trained models

ViT-based pre-trained models are fine-tuned using the AdamW optimizer with an initial learning rate of 1e-4 and a weight decay of 0.05 to mitigate overfitting. Training is conducted for 10 epochs on input images uniformly resized to 224×224. We employ standard data augmentation techniques, including random cropping and horizontal flipping.

## Broader Impact

The core contribution of our work is the theoretical and empirical demonstration that Vision Transformers (ViTs) lack an inductive bias for local texture features. This insight explains their limitations in fine-grained texture recognition and underscores the need to embed biases such as locality and translation equivariance into their design. By analyzing texture bias and its causes, we aim to guide the development of ViT architectures and training strategies that address these shortcomings.

Incorporating such biases would significantly improve ViT performance in fine-grained tasks, especially in data-scarce settings where capturing local detail is critical. Better understanding of training objectives, data augmentation, and architectural adaptations can enable ViTs to generalize more effectively, which is vital for real-world applications like medical diagnosis, where missing fine textures can affect

outcomes. Enhanced texture recognition also benefits industries such as manufacturing and surveillance, where improved precision reduces costs and boosts automation.

Ultimately, tackling texture bias in ViTs has broader implications for advancing general AI, promoting fairness by reducing model biases, and increasing economic efficiency in AI-driven systems. Our findings highlight the importance of continued research to make ViT-based foundation models more robust, interpretable, and suited for high-stakes deployment.