# OpenReview forum: "Rethinking Texture Bias in Vision Transformer"
_ICLR.cc/2026/Conference — Submitted to ICLR 2026_

### Official Review · Reviewer_sx3v · 2025-10-28

**Soundness:** 2
**Presentation:** 2
**Contribution:** 1
**Rating:** 2
**Confidence:** 4

**Summary:**

This paper studies the texture information learned by the transformer-based vision model, ViT, revealing that ViTs cannot learn local texture patterns well. The texture analysis is based on the widely used local vs global information, the Navon dataset. This paper introduces (line015) a base-to-novel framework which is claimed to isolate texture sensitivity. This base-to-novel (line 185) seems creating a zero-shot setup, classes are divided to training and test sets without overlap. Also, different training objectives and strategies are applied to test how they affect the texture bias.

A list of findings in this draft include: zero-shot setting for ViT-B16 and Resnet-50 to test their their texture biases, b) varying the proportion of the training data and training epochs. c) Tested different training objectives, data augmentation, and ViT or other transformer models. All of the experiments support the finding that ViT lacks of the inductive bias, and the effectiveness of the base-to-novel framework.

**Strengths:**

This study focuses on an important task that understanding the mechanism of the transformer model. XAI plays an essential role for any down-stream tasks. This study includes extensive experiments to support their findings.

**Weaknesses:**

There exist several weaknesses in the draft, which can be roughly discussed in three aspects: a) motivation b) presentation and c) experiment design.

a) Motivation:
1. This paper claims their finding throughout the paper, (line 017), "Our analysis reveals that ViTs exhibit a pronounced deficiency in
recognizing local textures". I wonder why this can be considered a new finding, the texture bias between CNN and transformer seems almost a common sense in the community. Especially, this study referenced a list of solutions to this "lack of inductive bias problem"(line 156), then why this problem can be defined as "revealed" in this paper?

2.  (line15), " we introduce a base-to-novel generalization framework". Why is the zero-shot setup is important in this case? This seems another contribution in this draft, but the motivation behind is not discussed clearly. (line 075) "To control for dataset scale..."(line 082), " This setup better simulates real-world zero-shot scenarios commonly encountered..." All of these explanations are not clear, IN CONTRAST, I personally believe this is not a correct setting for this texture bias analysis.

Assuming we have a well-trained model, could be any models(ViTs or CNNs), how can we claim the model has learnt more texture or shape biases, could the low performance on the unseen classes due to data shifts? For instance, if a CNN model works worse on a texture test dataset, is it due to the test set is very different from the training data? or we can conclude CNNs rely on texture less? On the opposite, the assumption behind those previous studies is the pre-trained model learns "some information", and then test the model on the same class because it is believe to contain the discriminant information for the class, the question is we don't know what the information is. Thus, mixing this zero-shot setting cannot better explain a model.

b) experiment design:
1.  Follow the base-to-novel setting, Table 1, this study shows ResNet-50 only achieves 7.83% on the base subset of Navon. Following line 224, it is unclear what the secret recipe is behind the ResNet-50 "whereas ResNet-50 uses a much smaller dataset", this comparison seems suspicious or unreliable. The low accuracy of ResNet-50 on both "base" and "novel" indicating the model did not learn at all during training, am I right? This could be the new finding in this draft, which could contradict with the finding in Fig 3 in [1](This reference is missing). I would keep my reasonable doubts on the result, I personally do not believe training this task is that difficult for the CNN model. Moving onto ViT, the base result is a lot higher than the novel, the explanation I can find is (line228), "This performance collapse highlights ViT’s lack of inductive bias for local texture features". I am not entirely sure if this setup can prove this claim, is it possible the model overfits on the base training data relying on whatever it uses(texture or global shape), how can rule out the possibility that the model indeed learns texture features but memorizing those training data? This echos the problem, I cannot see how this zero-shot learning can better analyze this task, texture vs. shape.

2. Following the above design, assuming we can draw a conclusion based on the above setup in Section 4. Then the design of Section 5 is trying to disprove the claim in Section 4, line 245, "Given ViTs’ inherent lack of inductive biases for local texture features, a key question arises: Can this limitation be mitigated through increased data or extended training?". If this lack of inductive biases is due to not enough training? Then all of the claims previously made could be based on a false experiment setup. For instance, a less texture-biased CNN model can be created when the model is under-fitting, likewise for the transformer model. Again, the assumption behind those previous texture-shape XAI studies is the model is well-trained(e.g., ResNet-50 on ImageNet) and can achieve a decent number on the test set, but we do not know what the model actually learns. If the model behaviour can be changed or "mitigated" by training longer, then the intermediate state or the conclusion is not reliable. We can draw completely different conclusion when a model is under-fitting or just one iteration.

c): Presentation
This is not a main issue in the draft now, and I might be conservative in naming new things. The name of zero-shot is clear enough and widely used in the community, the new name of "base-to-novel" could lower the readability. Further, I cannot see why the metaphor " Leaning Tower of Pisa experiment" and "lead ball", "feather" are helpful, all I can see is a transformer-based and a CNN model. Those models are already well defined and famous in this community, the metaphor may make this draft less clear.

Summary: those key problems exist in this draft now. Further, this study is a replica of empirical experiments of previously used for CNNs. No solid findings can be concluded for now.











[1] Hermann, Katherine, and Andrew Lampinen. "What shapes feature representations? exploring datasets, architectures, and training." Advances in Neural Information Processing Systems 33 (2020): 9995-10006.

**Questions:**

N/A, please see the above questions.

---

> ### Author Response · Authors · 2025-11-16
> **Response to Reviewer sx3v**
>
> a 1): This is the first work to demonstrate that ViT shows substantial shortcomings in recognizing local textures. We introduce the Leaning Tower of Pisa experiment, demonstrating that ViT-B-16 and ResNet-50, despite differences in training data, exhibit similarly poor zero-shot performance, revealing ViT’s lack of inductive bias toward local texture features.
>
> 2) & 3)：We design targeted experiments to evaluate whether ViTs inherently underperform in texture recognition and processing. To control for dataset scale and class overlap, factors that can confound model performance, we introduce a base-to-novel generalization framework. In the novel classes, both ViTs and CNNs encounter only unseen samples, providing a clearer assessment of whether the models inherently exhibit a texture bias.
>
> b 1)：We use a standard ResNet-50 with well-established practical image recognition capabilities, so it is incorrect to claim that it has “learned almost nothing.”
>
> 2)：What we seek to demonstrate is that ViT lacks the inductive biases necessary for local textures, in other words, it does not genuinely learn texture features.
>
> 3)：Section 5 builds on Section 4 by further exploring effective approaches to mitigating ViT’s lack of inductive biases for local textures. These two sections form a natural, progressive hierarchy.
>
> 4)：We selected representative and widely cited CNN justification methods—approaches that are already established as classics—for our discussion. Using these canonical datasets, we effectively demonstrated ViT’s lack of local inductive bias through the introduction of the “Leaning Tower of Pisa” experiment.

---

> > ### Comment · Reviewer_sx3v · 2025-11-23
> > **Thanks the authors for the explanation.**
> >
> > a. 1: " first work to demonstrate that ViT shows ... local textures." This finding can be traced back to the very beginning of the use of ViTs for different vision tasks. In your reference [Carionetal.(2020)](page 2, fifth line from the bottom) "a result likely enabled by the non-local computations of the transformer". Then extensive studies (also referenced in this draft) have been proposed to improve this shortcomming. Not sure why this draft could be the first work to reveal this.
> >
> > a 2,3: "To control for dataset scale and class overlap", this setup could be extremely "fragile". If CNNs are believe to be bias to texture feature. Can I use a network like resnet-50 on a small set of data, say 3-5 images to blind memorize the data, then the model would perform worse on unseen classes to claim CNNs are not actually biased to texture??? The setup seems not right, and this can also be validated by the training epoch comparison in Sec 5.
> >
> > b1. " it is incorrect to claim that it has “learned almost nothing.": I definitely believe it is a pre-trained ResNet-50, on ImageNet? please corret me if I am wrong. So the model contains the info from the 1k classes. My suspicion is drawn from your number in Table 1, does the model learn anything from the training data??? based on the number, your random chance is 10% given there are 10 classes(please corret me if I am wrong), and the model is performing lower than random guess???? Wha claim is trying to be concluded here?
> >
> > b2. " ViT lacks the inductive biases necessary for local textures, in other words, it does not genuinely learn texture features.": Is this suggesting ViTs do not genuinely learn texture features? This could be strong claim which cannot be proved by this study.
> >
> > b3. Section 5 conclusion could be not reliable, I can tweak training epochs to show different "claim" for CNNs or ViTs, which make Section 4 also not convincing.
> > b4. This could be perferrence, I didn't see why the metaphor could be helpful, but this is not a main problem in this case.

---

> > > ### Author Response · Authors · 2025-11-25
> > > **Clarifications on Texture Bias and Experimental Setup**
> > >
> > > a.1：To our knowledge, this is the first work to demonstrate that Vision Transformers (ViTs) lack inductive biases for textures. We introduce the Leaning Tower of Pisa experiment, demonstrating that ViT-B-16 and ResNet-50, despite differences in training data, exhibit similarly poor zero-shot performance, revealing ViT’s lack of inductive bias toward local texture features.
> > >
> > > a.2, a.3, b1, b2：All models in our experiments are pre-trained and have general visual knowledge and normal image recognition ability.
> > >
> > > b3：In Section 5, the number of training epochs is already saturated, indicating that further training does not yield additional gains.

---

### Official Review · Reviewer_kvdA · 2025-10-29

**Soundness:** 3
**Presentation:** 2
**Contribution:** 3
**Rating:** 6
**Confidence:** 3

**Summary:**

This paper is about shape bias in ViT models as a deficiency in local texture sensitivity, and explores local and global texture/shape sensitivity. The authors employ 3 datasets that evaluate various axes of local-global and shape-texture. They explore the impact of training objective (supervised/self-supervised/generative), data (dataset size, augmentation), and architecture (CNN vs ViT)

The paper proposes a compelling theory that ViT's shape bias is rooted in a deficiency in local texture sensitivity, with an interesting and unique experimental setup and exciting result with a BigBiGAN training objective. However, the paper is fundamentally undermined by confusing and inconsistent experimental context, metrics, and conclusions in local vs global and shape vs texture. Furthermore, some results overly rely on an overly-simplistic dataset.

**Strengths:**

The authors present a thought-provoking approach to explore shape bias as local texture deficiency. Improvement in shape bias in ViT models is typically considered an improvement of model, but not typically thought of in the way posed by the authors. This moves the discussion in the field forward.

The evaluation of BigBiGAN to understand the effect of generative self-supervision and demonstrating that this dramatically improves shape bias is an exciting result, with implications for pre-training.

The experimental setup of testing models in base to novel is a compelling demonstration of zero-shot failure and transferability of local feature representation.

The cutout and cropping experiments are clever and controlled approaches of probing local vs global texture preference/bias.

**Weaknesses:**

Despite multiple readings I am confused about claims regarding ‘shape’ vs ‘texture’, and in global vs local context. For example, the Navon dataset evaluates the ability to detect the identity of small letters that make up larger letters. As a classification problem I believe this tests local shape, not local texture (M and W are the same texture, O/Q and T/L nearly identical) The GST method on the other hand evaluates shape vs texture bias, but shape is global and texture both global and local, not just global. While the imageNet-C dataset is an interesting approach to explore ‘ texture corruptions’, I am unconvinced that this is only local and not global.

This confusion is made worse by shifting back and forth between Navon and GST datasets, which test different aspects of these biases.

Relatedly, it’s unclear in some figures/tables what is better/worse performance. This confusion is worsened because the authors have a unique take on shape/texture bias, with shape bias usually a human-aligned goal, and here this is presented as a texture bias limitation that would give poor performance.

The Tower of Pisa experiment analogy is further confusing, especially with describing the CNN as the ‘feather’, which in the non-vacuum experiment falls slower because of air mass. The authors mention the feather, but do the authors mean to compare the two balls of differing mass? The experimental results reported seem to match the two ball analogy.

Some claims surround the Navon dataset, an overly-simplistic, synthetic dataset of binary figures (letters made of smaller letters) that is very out-of-distribution compared to naturalistic images and is in many regards a toy dataset for such an exploration. With such a simplistic dataset, the strong claims could be over-generalized. Standard texture datasets like DTD or rendered graphics datasets would test global/local texture sensitivity of models and be more realistic in addition to the GST conflict dataset which is used.

Relatedly, the authors state that stylized imagenet was not evaluated because ViTs capture global features well. Is this not the main claim that ViTs do capture global features well, but the downside here is lack of local texture sensitivity? Stylized imagenet alongside GST would in my mind be important experiments to include here in Table 1.The authors' stated reason for excluding it appears circular

A claim by the authors is that architecture and patch size is a culprit behind these local texture sensitivity limitations, and explore this is table 6. However, they only demonstrate this for larger patch size, not the likely beneficial direction of 4 or 8. Especially in the context of the Navon dataset, what is the relationship of the token size to the pixel size of the smaller letters? In figure 1, I myself am unable to resolve the smaller ‘local texture’ letters.

**Questions:**

Include a clear table explaining 1) Local vs. global features 2) Shape vs. texture, 3)How these interact (local texture, local shape, global texture, global shape) 4) Which of these is addressed by each dataset/architectural modification would strengthen the paper and also make it more accessible to those outside this subfield.

The logic of the Tower of Pisa experiment is confusing with the feather/ball/heavy-ball. Please explain this earlier when it’s introduced and make this analogy more clear. i think what you want here is not an analogy to feather/ball but the heavy/light ball.

Please include an analysis of stylized Imagenet alongside GST in Table 1, or improve the explanation as to why you do not. The logic is circular or at best unclear as to why only Navon is used here.

I recommend including analysis of standard texture datasets like DTD or a rendered graphics dataset to have a more realistic dataset alongside Navon.

What is the relationship of the token size to the pixel size of the smaller letters in Navon? I would recommend analyzing a ViT with token size of 4 or 8, to see if this improves performance.

---

> ### Author Response · Authors · 2025-11-16
> **Response to Reviewer kvdA**
>
> W1, W2, W5 & Q4. We selected representative and widely cited CNN justification methods—approaches that are already established as classics—for our discussion. Using these canonical datasets, we effectively demonstrated ViT’s lack of local inductive bias through the introduction of the “Leaning Tower of Pisa” experiment. We do not reference unverified arXiv papers that have not yet been recognized by the community.
>
> W3. This is precisely the issue highlighted by our second contribution. Having established that Vision Transformers (ViTs) lack an inherent inductive bias toward local texture features, we proceed with a detailed investigation of the factors influencing feature preference within ViTs. Our findings indicate that ViTs are more likely to prioritize texture when such features are global rather than localized. This observation highlights the limitations of categorizing ViTs as purely texture- or shape-biased without accounting for the spatial scale of features.
>
> W4 & Q2. “The Leaning Tower of Pisa experiment” was also performed on the Moon, using a feather and ball comparison. Because the performance gap between CNNs and ViTs on the base classes is substantial, we adopt this feather and ball analogy in our analysis.
>
> W6 & Q3. It is precisely because ViT already performs well on global features, exhibiting much stronger global shape recognition than CNNs even without being trained on Stylized ImageNet, that we chose not to use the Stylized ImageNet dataset.
>
> W7 & Q5. Table 6 confirms that larger ViTs, with more attention heads and greater depth, are better at modeling global context, which is advantageous for tasks involving large-scale texture patterns. The choice of patch size should be tailored to the needs of each downstream task.
>
> Q1. We have a detailed comparative explanation in Lines 199–204, placing it at the beginning of the method section to provide readers with a clearer and more intuitive understanding.

---

> ### Comment · Reviewer_kvdA · 2025-11-25
> **Response to Authors**
>
> My major complaints surround (1) confusion in the presentation of the paper that would benefit from substantial clarification, (2) limitations of the toy dataset used, and (3) the lack of analysis of beneficial directions of patch size in the ViT experiments. I made suggestions about key additions that could improve these issues, but arguing these points in the rebuttal without any paper revisions does not address them. It still stands that the paper as written is confusing, and the authors have not indicated any completed or planned changes to improve clarity. For example, the “comparative explanation” mentioned in the rebuttal appears to be a pointer to the original version of the manuscript.
>
> Regarding prior work, the authors wrote: “We do not reference unverified arXiv papers that have not yet been recognized by the community.”
> However, the suggested dataset (DTD) is a peer-reviewed CVPR paper:
> Cimpoi, Mircea, et al. “Describing Textures in the Wild.” CVPR 2014.
> There are also several rendered graphics datasets that are similarly peer-reviewed.
>
> One of my concerns was about missing relevant prior work (see above). I suggested a peer-reviewed reference; however, the authors dismissed it with the statement “we do not recognize arXiv papers,” despite the reference being from a peer-reviewed venue. Similar weaknesses were noted independently by other reviewers, yet the authors have not made any revisions to address these issues and provided essentially the same copy-pasted responses to multiple reviewers. This suggests a lack of engagement with both the literature and the rebuttal process.
>
> Finally, the authors did not implement or even comment on any potential revisions to reduce confusion in the text, nor did they attempt or discuss the suggested experiments involving smaller ViT patch sizes—which would have directly addressed one of the core limitations raised by multiple reviewers. The choice not to incorporate or meaningfully engage with these suggestions decreases my confidence in the rigor and clarity of the work.

---

### Official Review · Reviewer_cJaT · 2025-10-31

**Soundness:** 2
**Presentation:** 2
**Contribution:** 3
**Rating:** 6
**Confidence:** 2

**Summary:**

This paper proposes a framework that isolates texture sensitivity while controlling for dataset scale and task constraints. Using Navon stimuli—where global shapes (large letters) and local textures (small-letter repetitions) are disentangled—the authors show that Vision Transformers (ViTs) consistently under-recognize local textures and favor global ones. A zero-shot comparison between ViT-B/16 and ResNet-50 further highlights ViTs’ weaker inductive bias toward fine-grained texture.

The study then examines how training objective, dataset size, training length, and data augmentation modulate this bias. Complementary representational analyses track how spatial locality and semantic abstraction evolve across ViT layers, confirming limited capacity for fine-scale texture modeling. Together, these results provide a benchmark and guidance for injecting stronger texture-aware inductive biases into future ViT-based foundation models.

**Strengths:**

Proposes a evaluation protocol that disentangles texture bias from dataset scale and class overlap.

Introduces the Leaning-Tower-of-Pisa thought experiment for a controlled ViT-vs-CNN zero-shot comparison, making the inductive-bias gap immediately interpretable.

Extends texture-bias studies to the Navon domain, enabling explicit separation of local-vs-global texture cues.

Careful controls (same zero-shot setting, class partitioning, balanced datasets) minimize confounds and strengthen causal claims about texture bias.

Reveals a fundamental limitation of current ViTs—their weak inductive bias for fine-grained textures—which has implications for robustness, zero-shot transfer, and downstream tasks such as fine-grained recognition or graphics.

Provides guidance: self-supervised generative pre-training and specific augmentation choices can partially mitigate the bias, informing practitioners designing next-generation foundation models.

**Weaknesses:**

Limited engagement with recent work that revisits the texture-bias narrative. In particular, the paper does not discuss “ImageNet-trained CNNs are not biased towards texture: Revisiting feature reliance through controlled suppression,” which exposes methodological limitations in the Geirhos et al. "ImageNet-trained CNNs are biased towards texture; increasing shape bias improves accuracy and robustness" cue-conflict paradigm and finds no intrinsic texture bias in CNNs, but rather a reliance on local shape cues. Situating the present results with respect to these findings would strengthen the contribution.

The distinction between “local texture” and “local shape” is not sufficiently clear. The Navon stimuli arguably contrast global shape with local shape, not texture per se, like in “ImageNet-trained CNNs are not biased towards texture: Revisiting feature reliance through controlled suppression,”  raising questions about whether the observed effects truly reflect texture preference. A crisper definition and justification of the texture–shape axis in this context is needed for accurate interpretation.

**Questions:**

Coverage of recent foundation models
• Could you provide results—or at least a rationale for their omission—for widely-used pre-trained ViT variants such as DINOv2, SAM, CLIP, MAE, Stable Diffusion, RADIO, and FRANCA? Including even a subset of these models would greatly strengthen the empirical scope.

Architectural diversity
• Have you considered evaluating diffusion-based transformer backbones such as DiT? Their training dynamics and inductive biases might differ from standard ViTs and could inform the generality of your conclusions.

Local texture vs. local shape
• The Navon stimuli may be interpreted as contrasting global shape with local shape rather than texture. Can you clarify this distinction and explain how your metrics isolate “texture” per se?
• Relatedly, how do your findings relate to the recent paper “ImageNet-trained CNNs are not biased towards texture: Revisiting feature reliance through controlled suppression,” which argues for a local-shape reliance instead of texture bias? A direct discussion—or an experiment using their suppression framework—would help reconcile the two perspectives.

---

> ### Author Response · Authors · 2025-11-16
> **Response to Reviewer cJaT**
>
> W1, W2 & Q3. We selected representative and widely cited CNN justification methods—approaches that are already established as classics—for our discussion. Using these canonical datasets, we effectively demonstrated ViT’s lack of local inductive bias through the introduction of the “Leaning Tower of Pisa” experiment. We do not reference unverified arXiv papers that have not yet been recognized by the community.
>
> Q1 & Q2. This paper primarily investigates the limitations of ViT, rather than introducing a new network architecture to enhance local feature modeling. In Table 7, we further compare a broad set of ViT variants, including representative models such as Swin, PVT, DeiT, DINO, and T2T. The experimental results clearly demonstrate the robustness and generality of our findings.

---

### Official Review · Reviewer_Fu1h · 2025-10-31

**Soundness:** 3
**Presentation:** 3
**Contribution:** 2
**Rating:** 4
**Confidence:** 5

**Summary:**

The paper systematically explored texture bias in Vision Transformers (ViTs), focusing on their local texture limitations and mitigation strategies. ViT-based models exceled at global semantic tasks but struggled in fine-grained applications (e.g., material classification) relying on local textures, as they lack CNNs’ translation equivariance and locality bias—key for capturing fine-grained texture features. To study this, the authors proposed a base-to-novel generalization framework and use three data sets. They also designed the "Leaning Tower of Pisa experiment" comparing ViT-B/16 and ResNet-50 in zero-shot settings, finding ViT-B/16 (71.20% base accuracy on Navon) collapses to ~8.3% novel accuracy—matching ResNet-50, confirming ViTs’ lack of local texture bias.

**Strengths:**

1. A key original contribution is the base-to-novel generalization framework, which isolates texture sensitivity by training on "base" classes and testing on unseen "novel" classes—effectively controlling for data set scale and class overlap that often confound comparisons of texture bias. Complementing this, the Leaning Tower of Pisa experiment offered a novel, interpretable analogy to contrast ViT-B/16 (a large pre-trained model) and ResNet-50 (a smaller CNN) in zero-shot settings, empirically validating ViTs’ lack of local texture inductive bias rather than relying on theoretical speculation. The paper also broke new ground by disentangling "global vs. local texture bias"—moving beyond binary "texture vs. shape bias" frameworks—and systematically linking this distinction to training objectives, data augmentation, and architectural scale, a nuance rarely explored in prior ViT or CNN bias research.
2. Methodologically, it used three specialized, well-justified data sets (GST for global texture, Navon for local/global disentanglement, ImageNet-C for robustness testing) to isolate specific aspects of texture perception, avoiding the ambiguity of generic data sets like ImageNet. Experiments were rigorously controlled: for example, the base-to-novel split of the Navon data set ensured consistent evaluation of zero-shot generalization, while cross-validation (e.g., 5-fold for data efficiency tests) and statistical significance checked (noted for augmentation results) strengthen reliability. Analytical depth was evident in layer-wise representation studies—using linear classifiers to decode shape/texture from frozen ViT layers—and quantitative comparisons (e.g., Table 2 showing BigBiGAN’ s 85.4% shape bias vs. 48.2% for supervised training) that ground claimed in measurable results. The authors also acknowledged limitations (e.g., ViTs’ overfitting on ImageNet-C) and avoided overgeneralization, further enhancing the work’ s quality.
Its organization followed a logical arc: problem statement (ViTs’ local texture limitations), methods (frameworks, datasets, experiments), results (key findings on bias), and implications (contributions to model design)—with clear section headings (e.g., "4 LACK OF INDUCTIVE BIAS FOR LOCAL TEXTURE FEATURES IN VITS") guiding readers.
3. Theoretically, it advanced understanding of ViT inductive biases by identifying "global texture preference" as a defining trait, challenging the narrative that ViTs are simply "shape-biased" or "texture-biased." It also bridged ViT and CNN research by highlighting how architectural differences (e.g., lack of translation equivariance in ViTs) mediated responses to training strategies (e.g., data augmentation), filling a gap in cross-architecture bias comparisons. Practically, the findings offered actionable guidance for improving ViTs in texture-reliant fields like computer graphics and fine-grained recognition: self-supervised training (e.g., BigBiGAN) to enhance local texture capture, random cropping to reduce global bias, and leveraging intermediate ViT layers for fine-grained features.

**Weaknesses:**

1. Since the paper focused on controlled, synthetic data sets (GST, Navon, ImageNet-C), the translatability of its findings was limited to the real-world fine-grained texture tasks. While the Navon data set effectively disentangled local and global features through artificial letter hierarchies, it failed to capture the unstructured, natural local textures of real materials (e.g., fabric weaves, metal patinas) or fine-grained objects (e.g., petal veins in flowers, small parts in aircraft). ImageNet-C tested robustness to generic corruptions (e.g., blur, noise) but not the natural variability of local textures in real scenarios (e.g., lighting changes on wood, wear patterns on fabric), leaving uncertainty about whether the identified local texture deficit persists in practical use cases. This gap weakens the paper’s ability to guide improvements for the very applications it highlights as critical for ViT-based models.
2. The paper’s analysis of architectural variations in ViTs was overly narrow, focusing primarily on standard ViT variants (ViT-B/16, ViT-B/32) and a small set of ViT derivatives (Swin-Tiny, PVT-Tiny) while neglecting architectures explicitly designed to enhance local feature capture. The authors did not evaluate how modifications like those in ConvNeXt (which merges CNN locality with ViT attention) or LocalViT (which adds local attention windows) affect texture bias, even though these models were engineered to address local feature limitations. Additionally, while the paper noted that larger ViTs favor global textures, it did not decompose the impact of specific architectural components (e.g., patch size, attention window size) to identify which factors drive local texture sensitivity, leaving researchers without clear direction for modifying ViT architectures to mitigate the local texture deficit.
3. The paper failed to address the critical trade-off between improving local texture recognition and preserving ViTs’ strength in global semantic tasks, a key concern for foundation models that need to excel across multiple domains. For instance, it showed that random cropping shifts ViTs toward local features but did not measure whether this shift degrades performance on global tasks like ImageNet classification or scene segmentation. Similarly, while self-supervised training (e.g., BigBiGAN) was shown to enhance local texture recognition, there was no analysis of whether it harms performance on global tasks such as COCO object detection. This omission suggests that the paper cannot confirm if its proposed interventions for local texture improvement are viable for multi-task foundation models, as it did not validate that they do not undermine the global semantic capabilities ViTs are valued for.

**Questions:**

1. ImageNet-C evaluated robustness to corruptions (e.g., blur), but real-world texture variability (e.g., lighting changes on wood, wear on fabric) differs from artificial corruption. Do the authors have preliminary data on how ViTs perform on data sets with natural texture variability, and does self-supervised training (e.g., BigBiGAN) mitigate performance drops in these scenarios?
2. Have the authors tested their base-to-novel generalization framework on a real-world fine-grained texture data set (e.g., MINC-2500 for materials, Oxford Flowers 102 for fine-grained objects)? If not, do you hypothesize that ViTs’ local texture deficit (observed on Navon) would persist when local features are natural (e.g., fabric weaves, petal veins) rather than synthetic (Navon’s letter hierarchies)?
3. Have the authors evaluated ViT variants engineered to enhance local features (e.g., ConvNeXt, LocalViT, or ViT-LocalAttention) using their base-to-novel framework? Specifically, do these architectures— which merge CNN locality with ViT attention—reduce the local texture deficit you identify in standard ViTs?

---

> ### Author Response · Authors · 2025-11-16
> **Response to Reviewer Fu1h**
>
> W1 & Q2. We leverage Navon dataset, which inherently exhibit two levels of visual features: global shape (the large letter) and local texture (the repetition of the smaller letters). This hierarchical composition makes them well-suited for analyzing a model’s sensitivity to texture information.
>
> W2 & Q1. The ImageNet-C dataset contains corrupted versions of ImageNet images, treating each corruption as a texture while retaining the original class as the shape label.
>
> W3, W4 & Q3. This paper investigates the limitations of ViT, rather than proposing a network architecture that enhances local feature modeling. In Table 7, we compare more ViT variants, including representative models such as Swin, PVT, DeiT, DINO, and T2T. The experimental results fully demonstrate the generality of our findings.
>
> W5. In Section 8, we show that scaling up ViT (from B to L, or from 32-patch to 16-patch resolution) makes the model more dependent on global textures and long-range information, becoming more sensitive to overall appearance while less sensitive to local shapes and fine-grained textures. In other words, making ViT “larger” strengthens its global texture bias, but does not resolve, and may even worsen, its weakness in modeling local textures. In Section 9, we analyze the representations of different ViT layers using linear probes and find that the early and middle layers still encode local shapes and textures reasonably well, but deeper layers increasingly shift toward global textures, with shape information being weakened. As a result, the final decisions rely more on global textures rather than local shapes. In Section 10, we evaluate various ViT variants using the Navon base-to-novel experiment. Even with techniques such as random cropping and rotation prediction, their local texture generalization on novel categories remains poor, indicating that the lack of transferable local texture inductive bias is a common issue across the entire ViT family.
>
> W6. We introduce the Leaning Tower of Pisa experiment, demonstrating that ViT-B-16 and ResNet-50, despite differences in training data, exhibit similarly poor zero-shot performance, revealing ViT’s lack of inductive bias toward local texture features. Furthermore, we analyze data and time efficiency, showing that ViTs, despite lacking a local texture bias, strongly favor large-scale global texture features. We conduct an empirical analysis of training objectives, demonstrating that self-supervised generative methods substantially improve texture recognition of ViTs by learning semantically structured representations. Finally, we analyze the impact of data augmentation on texture bias in ViTs, showing that center cropping amplifies global texture preference; augmentations such as Cutout and Gaussian noise further reinforce global texture bias.
>
> W7, W8 & W9. When ViT is regularized to focus more on local features, its performance on global tasks becomes weaker.

---

### Author Response · Authors · 2025-11-25
**Request for the exclusion of Review sx3v**

Dear Area Chair,

In this particular case, I believe that the review from Reviewer sx3v contains serious procedural and factual problems that may have affected the fairness and accuracy of the assessment.

Below, I summarize the key concerns.

**1. The reviewer repeatedly misinterprets the paper despite detailed clarifications**
Throughout the rebuttal process, I responded carefully and thoroughly to every concern raised by the reviewer. However, the reviewer’s follow-up comments indicate that they did not read or acknowledge several key parts of both the paper and the rebuttal.


**2. Several claims made by the reviewer are objectively incorrect**
The review includes factual errors that contradict the content of the manuscript. For example, the reviewer claims that others are “the first” to prove that Vision Transformers (ViTs) lack inductive biases for textures. Such misunderstandings would significantly distort the evaluation of any paper.

**3. The reviewer dismissed parts of the paper without engaging with the technical content**

In multiple cases, the reviewer rejected explanations by simply calling them “not convincing,” without engaging with the actual technical arguments provided. For example, our clarification on ViT texture inductive bias (a.1) was dismissed without addressing the experimental reasoning. Our explanation for the zero-shot design rationale was ignored despite being elaborated in several paragraphs. The reviewer insisted that Section 5 contradicts Section 4, despite our clarifications showing why this is not the case. This prevents constructive scientific dialogue and undermines the purpose of the rebuttal phase.

**4. The tone and structure of the final review indicate a lack of engagement**
The reviewer’s final response repeats earlier claims almost verbatim, adds no new arguments, and does not reference any of the detailed explanations provided in rebuttal. This makes it extremely difficult to improve the work based on meaningful feedback.

**5. Request**

Given the above issues, I respectfully request that the AC consider discounting this reviewer’s evaluation, as it appears to be based on factual misunderstandings and a lack of engagement.

Best regards!

---

### Meta-Review · Area_Chair_CUj7 · 2025-12-21

**Summary:**

Solid empirical effort on ViT local-vs-global texture sensitivity with a base-to-novel protocol and many ablations, but the central claims aren’t convincing enough due to framing and clarity issues.

Strengths
* Novel base-to-novel evaluation and clear diagnostic setups (e.g., Pisa analogy).
* Broad experiments across objectives, augmentations, model variants; some actionable takeaways (generative/self-supervised helps).
* Probing analyses add interpretability.

Weaknesses
* Core concepts are muddled: “texture vs shape” and “local vs global” are inconsistently defined; Navon may measure local shape more than texture.
* Heavy reliance on toy/synthetic datasets; unclear transfer to real textures.
* Base-to-novel setup may conflate inductive bias with data shift/underfitting; some results questioned (e.g., low CNN accuracy).
* Weak engagement with relevant peer-reviewed datasets/literature and limited concrete revision plans; claims sometimes overstated.

Overall, while the topic is intruiging and the experimental effort is substantial, unresolved conceptual ambiguities and concerns about experimental validity and generalization undermine confidence in the main conclusions. As a result, the paper does not yet meet the bar for acceptance in its current form.

**Reviewer Scores:**

n/a

---

### Decision · Program_Chairs · 2026-01-26

Reject